# Colorectal Cancer Stage-Specific Fecal Bacterial Community Fingerprinting of the Taiwanese Population and Underpinning of Potential Taxonomic Biomarkers

**DOI:** 10.3390/microorganisms9081548

**Published:** 2021-07-21

**Authors:** Chuan-Yin Fang, Jung-Sheng Chen, Bing-Mu Hsu, Bashir Hussain, Jagat Rathod, Kuo-Hsin Lee

**Affiliations:** 1Division of Colon and Rectal Surgery, Ditmanson Medical Foundation Chia-Yi Christian Hospital, Chiayi 621, Taiwan; 04969@cych.org.tw; 2Department of Medical Research, E-Da Hospital, Kaohsiung 824, Taiwan; nicky071214@gmail.com; 3Department of Earth and Environmental Sciences, National Chung Cheng University, Chiayi 621, Taiwan; bashir.aku@gmail.com; 4Center for Innovative on Aging Society (CIRAS), National Chung Cheng University, Chiayi 621, Taiwan; 5Department of Biomedical Sciences, National Chung Cheng University, Chiayi 621, Taiwan; 6Department of Earth Sciences, National Cheng Kung University, Tainan 701, Taiwan; jagat2006@gmail.com; 7Department of Emergency Medicine, E-Da Hospital, I-Shou University, Kaohsiung 824, Taiwan; peter1055@gmail.com; 8School of Medicine, I-Shou University, Kaohsiung 824, Taiwan

**Keywords:** colorectal cancer, gut microbiota, gut microbial dysbiosis, prognosis, biomarker, metagenomics, functional predictions

## Abstract

Despite advances in the characterization of colorectal cancer (CRC), it still faces a poor prognosis. There is growing evidence that gut microbiota and their metabolites potentially contribute to the development of CRC. Thus, microbial dysbiosis and their metabolites associated with CRC, based on stool samples, may be used to advantage to provide an excellent opportunity to find possible biomarkers for the screening, early detection, prevention, and treatment of CRC. Using 16S rRNA amplicon sequencing coupled with statistical analysis, this study analyzed the cause–effect shift of the microbial taxa and their metabolites that was associated with the fecal gut microbiota of 17 healthy controls, 21 polyps patients, and 21 cancer patients. The microbial taxonomic shift analysis revealed striking differences among the healthy control, polyps and cancer groups. At the phylum level, Synergistetes was reduced significantly in the polyps group compared to the healthy control and cancer group. Additionally, at the genus level and in association with the cancer group, a total of 12 genera were highly enriched in abundance. In contrast, only *Oscillosprira* was significantly higher in abundance in the healthy control group. Comparisons of the polyps and cancer groups showed a total of 18 significantly enriched genera. Among them, 78% of the genera associated with the cancer group were in higher abundance, whereas the remaining genera showed a higher abundance in the polyps group. Additionally, the comparison of healthy control and polyp groups showed six significantly abundant genera. More than 66% of these genera showed a reduced abundance in the polyps group than in healthy controls, whereas the remaining genera were highly abundant in the polyps group. Based on tumor presence and absence, the abundance of *Olsenella* and *Lactobacillus* at the genus level was significantly reduced in the patient group compared to healthy controls. The significant microbial function prediction revealed an increase in the abundance of metabolites in the polyps and cancer groups compared to healthy controls. A correlation analysis revealed a higher contribution of *Dorea* in the predicted functions. This study showed dysbiosis of gut microbiota at the taxonomic level and their metabolic functions among healthy subjects and in two stages of colorectal cancer, including adenoma and adenocarcinoma, which might serve as potential biomarkers for the early diagnosis and treatment of CRC.

## 1. Introduction

Colorectal cancer (CRC), also known as colorectal adenocarcinoma, is considered the second most leading cause of cancer-related mortality and is responsible for more than 1.3 million new cases per annum [1,2]. Such a growing rate of incidence makes it the most prevalent and significant health issue on a global scale. Generally, CRC arises from certain epithelial cells of the large intestine following a series of genetic and epigenetic perturbations. Subsequently, normal epithelial cells transform into adenomas (benign neoplasms or polyps) followed by invasive carcinomas [3,4]. Several risk factors including dietary habits (less fiber/excessive red meat), smoking, obesity, alcohol consumption, and diabetes have been linked to the initiation and progression of CRC [4,5,6]. Additionally, in recent years there have been cumulative reports finding the possible envelopment of human gut microbiota in CRC carcinogenesis [7,8].

The human gastrointestinal tract, especially the colon, harbors a diversified microbial community ranging from 10^13^–10^14^ microorganisms [9]. These microbes are primarily dominated by bacteria as the core microbiota in the gut of healthy people, with obligate anaerobes, mainly *Bacteroides*, *Firmicutes*, *Proteobacteria Actinobacteria*, and *Verrucomicrobia* comprising 95% of gut microbiota [10,11]. The gut microbiota performs a variety of crucial functions that help the host in protection from pathogens, nutrient supply, immune modulation, and shaping the intestinal epithelium [10,12]. Previous studies have shown that various gut microbiota and their associated metabolites exhibit proinflammatory and procarcinogenic properties which ultimately exert a great impact on colorectal carcinogenesis [13,14,15].

After the advent of high throughput sequencing, the interest in the relationship between cancer and intestinal microbiota has rapidly grown in recent years. High throughput sequencing has made many efforts to unveil the characterization, association, and functional prediction of gut microbiota in healthy and diseased subjects [16,17]. There are substantial reports, both in animal and human models, that suggest any alterations in microbial community structures greatly influence the colon health condition, leading to various intestinal disorders such as inflammatory bowel disease, diabetes, obesity, and various type of cancer, particularly colorectal cancer [14,17,18]. Other reports of CRCs have suggested the existence of linkages between gut microbiota and their metabolites in tumor initiation and progression in a murine colitis-associated CRC model [18,19]. Additionally, several reports highlighted evidence of fecal microbiota associated with CRC as being tumorigenic in germ-free mice models [20,21].

Previous studies have revealed that the intestinal mucosa and fecal flora are different in patients with colorectal cancer [15,22]. In the stool of healthy people, a higher bacterial diversity has been observed as compared to the stool of patients with colorectal cancer [23]. However, it has been observed that the bacterial diversity on the mucosa close to cancer cells is relatively low in the healthy person, whereas increased bacterial diversity is noticed in colorectal cancer patients [17]. Additionally, the overall structure of lumen noncancerous and cancerous microbiota found that a lower diversity was exhibited in similar but cancerous tissue [8]. The relative abundance of dominant phyla including Firmicutes, Bacteroidetes, Proteobacteria, and Fusobacteria have been observed differently in both healthy control and CRC patients [24]. In particular, Fusobacteria and Proteobacteria were more abundant in cancer specimens, whereas a significantly higher abundance of Fusobacteria and Firmicutes were observed in the stools of a colorectal patient [18]. According to recent studies, *Peptosteptococcus*, *Clostridium*, *Prutella*, *Porphyromonas*, and *Bacteroides* are the key bacteria linked with colorectal cancer [1]. Additionally, some other bacteria are currently considered to be linked with colorectal cancer, such as *Streptococcus bovis*, a Gram-positive opportunistic pathogen having the ability to grow in a 40% bile environment and cause systemic infections in the human body [25]. Similarly, other common bacteria of the human gut, such as *Streptococcus gallolyticus* and *Helicobacter pylori*, are believed to be involved in colorectal cancer [26]. Moreover, certain strains of *Escherichia coli* producing a toxin such as cyclomodulin, are also believed to be involved in the carcinogenesis of CRC [7]. Additionally, mice infected with a certain strain of *E. coli* have displayed a marked increase in the number of visible colonic polyps as compared to controls [27]. Previous reports also highlighted a profound shift of bacterial groups, less common or more common, in healthy subjects and CRC [23,28,29].

Additionally, it has been reported that gut microbial-associated metabolites are not only helpful in the maintenance of human gut health, but they also play a key role in the development of CRC [28,30]. There are substantial reports that the general gut microbiota is associated with the initiation and progression of CRC through the production of carcinogens, cocarcinogens, or procarcinogen substances [29]. The direct interaction of gut microbes with the epithelial cells is limited due to the presence of mucosal barriers [31]. However, these microbes convert the complex chemicals provided by the host and dietary nutrients into a milieu metabolite which can be easily translocated across the mucosal barrier and play a role in tumorigenesis through multiple mechanisms, such as modifying signaling proteins [28,31]. In previous studies, it has been demonstrated that the elevated gut microbes derived from secondary bile acids, such as deoxycholic acid, promote the development and progression of CRC [32,33]. Contrarily, the decreased level of beneficial gut microbial-associated metabolites, such as butyrate, is also involved in mutagenesis [34]. Additionally, gut microbiota-associated reductive and hydrolytic enzymes also play roles in the occurrence of CRC [35].

Thus, taking advantage of the shift in microbial taxa and their metabolites in CRC patient stool samples may provide an excellent opportunity to find possible biomarkers for the screening, early detection, prevention, and treatment of CRC. Ultimately this will help in the reduction of CRC-associated death globally. Therefore, in this study, we performed amplicon sequencing of 16S rRNA genes based on high throughput sequencing to analyze the overall bacterial compositions and their functional predictions that were associated with the human gut microbiota of healthy subjects, polyps, and cancer patients to find potential biomarkers associated with CRC.

## 2. Materials and Methods

### 2.1. Characteristics of Patients and Healthy Participants

In this study, a total of 17 healthy subjects were selected as a control group. Subjects were aged 25–95 years old (both male and female) with no acute and chronic gastrointestinal diseases and no antibiotic administration history within a one-month period prior to sample collection. Similarly, 21 persons aged 25–95 years old (both male and female) with large intestine tumor symptoms were divided into polyps and bowel cancer patients. After radiography examination, patients with colorectal cancer were at a T and L level, based on the TNM classification standard as previously described [18]. After diagnosis, these patients had not been treated with chemotherapy and radiotherapy. Besides, for both the control and the experimental groups the following conditions were excluded: (1) obesity, pregnancy, high blood pressure, stroke, myocardial infarction, diabetes, hyperlipidemia, gout, long-term lying, patients with primary and secondary parathyroid hyperthyroidism; (2) drugs, alcohol, and drug abuse; (3) taking drugs (antibiotics, constipation drugs) or food (probiotics) in the preceding month that could affect the gastrointestinal tract. Sample collections were conducted according to the guidelines of the Declaration of Helsinki and approved by the Institutional Review Board of Ditmanson Chia—Yi Christian Hospital, Taiwan (CYCH-IRB No. 2019021 and date of approval: 11 April 2019).

### 2.2. Sample Collection and DNA Extraction

The fecal specimens of healthy subjects and patients with colorectal cancer and polyps were collected in a sterile stool box with cryopreservation at Chiayi Christian Hospital, Taiwan. Samples were then transported following the biosafety procedures and under controlled temperature conditions to the laboratory at National Chung Cheng University, Taiwan. Fecal gDNA was extracted from a 200 mg stool sample using a QIAamp DNA Stool Mini Kit (QIAGEN) following the manufacturer’s instructions. Additionally, a bead-beating step was performed using a previously designed protocol [36]. In brief, 250 µL of the stool sample was taken in a 2 mL sterilized tube holding 1.2 mL ASL lysis buffer along with 0.3 g sterile 0.1 mm zirconia beads (BioSpec, Bartlesville, OK, USA) and this was followed by vortex-mixing for 2 min. The samples were subjected to heating for 15 min at 95 °C and subsequently homogenized using the Qiagen TissueLyser II. After treatment with an InhibitEX Tablet, 350 µL of supernatant was shifted to another tube to perform the subsequent purification steps using a QIAcube system.

The purity and concentration of the extracted gDNA was determined using a Nanodrop 2000 spectrophotometer (Thermo Fisher Scientific Inc., Wilmington, DE, USA) at 230–280 nm. The quality of the gDNA was examined using gel electrophoresis (1.5% gel in Tris-acetate ethylenediaminetetraacetic acid buffer) at 110 V for 30 min. The DNA bands were visualized under ultraviolet light. The purified gDNA was stored at −20 °C for further analysis.

### 2.3. Sequencing, Library Construction, and 16S rRNA Amplicon Data Analysis

We amplified the V3–V4 region of the hypervariable region of the 16S rRNA gene from the extracted gDNA using 341F and 805R primer sets (with some modifications) with the Illumina adapter overhang sequence attached at the 5′ end of the primers. The sequences of the forward primers used in this experiment were as follows: (16S_341F) 5′-**TCGTCGGCAGCGTCAGATGTGTATAAGAGACAG**CCTACGGGNGGCWG CAG-3′, (16S_341F _N) 5′-**TCGTCGGCAGCGTCAGATGTGTATAAGAGAC** AGnCCTACGGGNGGCWGCAG-3′, (16S_341F _N N) 5′-**TCGTCGGCAGCGTCAG ATGTGTATAAGAGACAG**nnCCTACGGGNGGC WGCAG-3′, and (16S_341F _NNN) 5′-**TCGTCGGCAGCGTCAGATGTGTATAAGAGACAG**nnnCCTACGGGNGG CWGCAG-3′. The reverse primer sequences were as follows: (16S_805R) 5′-**GTCTCGTGGGCTCGGAGATGTGTATAAGAGACAG**GACTACHVGGGTA TCT AATCC-3′, (16S_805R _N_) 5′-**GTCTCGTGGGCTCGGAGATGTGTATAAGAGA CAG**nGACTACHVGGGTATCTAATCC-3′, (16S_805R _NN) 5′-**GTCTCGTGGGC TCGGAGATGTGTATAAGAGACAG**nnGACTACHVGGGTATCTAATCC-3′, and (16S_805R _NNN) 5′-**GTCTCGTGGGCTCGGAGATGTGTATAAGAGACAG** nnnGACTACHVGGGTATCTAATCC-3′. The amplification was performed in triplicate, following the previously reported method [37]. The optimal PCR conditions were as follows: 95 °C for 3 min, followed by 30 cycles of 95 °C for 30 s (denaturation), 55 °C for 30 s (annealing), 72 °C for 30 s (primer extension), and 72 °C for 5 min (elongation). The quantity and quality of amplified DNA were assessed using the standard quality checks mentioned above. Next, the amplicons (20 µL) from each sample were subjected to sequencing using the pair-end method with the MiSeq Illumina platform (Illumina Inc., San Diego, CA, USA) following the standard protocol at the National Yang-Ming University Genome Research Center, Taiwan. The DNA libraries were ligated with the sequencing adapters and index using the Nextera XT sample preparation kit (Illumina), following the manufacturer’s instructions. The sequence data containing reverse and forward reads were aligned using the CLC bio plate form (Genomic Workbench v.8.5) and the FASTA files were generated as described in our previous study [38]. The QIIME2 system was used for the sequence quality control and amplicon sequence variant (ASV)-based classification [39]. In brief, the quality of raw sequencing reads was assessed by FastQC. DADA2 was used for denoising and constructing ASVs. The denoising steps also included the truncation of the forward (250–280 bp) and reversed reads (180–260 bp) based upon the quality profile and amplicon length which was performed by using DADA2. The minimum and maximum number of quality count reads was 4500 and 37,432 per sample, respectively. The rarefaction was performed at 4300 read counts to estimate the bacterial diversity. Finally, the relative abundance of microbes at the phylum and genus levels in each sample was obtained using the QIIME2 view. Furthermore, the significant difference in the relative abundance at the phylum and genus level in each group was analyzed using statistical analysis of taxonomic and functional profiles (STAMP) software [40]. The significant relative abundance at the phylum and genus level was examined using a two-tailed White’s non-parametric *t*-test (*p* < 0.05). Additionally, the linear discriminant analysis effect size (LEfSe) was performed in Galaxy software (http://huttenhower.sph.harvard.edu/lefse/, accessed on 6 July 2021) following an LDA score > 2 and *p* < 0.05 to identify the differential abundance in microbiota among the experimental groups [41]. To know the significance among the clustering pattern in the ordination plot, a permutational multivariate analysis of variance (PERMANOVA) was performed using MicrobiomeAnalyst, based on the experimental groups [42].

### 2.4. Metagenomic Functional Prediction Based on 16S rRNA Gene Data

To examine the potential metabolic function of microorganisms in each sample belonging to healthy control, polyps, and CRC patients, the representative sequence and denoised ASV abundance tables were used. These tables were input into the phylogenetic investigation of communities reconstruction of Unobserved States (PICRUSt2) pipeline (https://github.com/picrust/picrust2, accessed on 29 December 2020) using KEGG (Kyoto Encyclopedia of Genes and Genomes). All ASVs with a nearest-sequenced taxon index (cutoff value > 2) were removed by default for the reliable annotation of metabolic functions using the KEGG reference database as previously described [43]. Finally, using White’s nonparametric *t*-test in STAMP software (*p* < 0.05) with 95% confidence intervals, the results of level 2 KEGG pathways were used to ascertain the significant shift in the bacterial community in each group. A Spearman correlation analysis was performed using IBM SPSS Statistics 24 (IBM, Armonk and North Castle, NY, USA) to evaluate the significant correlations between bacterial diversity and the potential functional prediction, considering *p*-values in the range 0.01–0.05.

## 3. Results

### 3.1. Gut Microbial Richness and Diversity with Respect to Health Conditions and Based on 16S rRNA Amplicon Sequencing

In order to study bacterial community composition in each stage of CRC and in healthy controls, we performed 16S rRNA amplicon sequencing targeting the V3 and V4 regions, which after quality filtering and chimera removal generated a total of 1919 ASVs, based on a 97% cutoff value using the reference database (Greengenes). As shown in Figure 1A, among these, compared to healthy controls (*n* = 621), the cancer group (*n* = 901) exhibited a higher ASV number, followed by the polyps group (*n* = 800). Additionally, 107 ASVs were common among the three experimental groups, while the majority were still unique to each experimental group (413 for healthy controls, 549 for the polyps group, and 661 for the cancer group), revealing, compared to healthy controls, a higher unique ASV diversity in the cancer group followed by the polyps group. Additionally, the richness and diversity of the gut microbial community in each experimental group was evaluated using Chao1 (Figure 1B), Simpson (Figure 1C) and Shannon (Figure 1D) alpha diversity indices, revealing a higher alpha diversity in the cancer group compared to the healthy controls, but the difference was not statistically significant. The beta diversity analysis based on a weighted UniFrac distance matrix showed a low variation in microbial diversity and as a result no clustering patterns were noted among the experimental groups (Figure 1E). Additionally, the significance analysis of clustering patterns in the ordination plot was evaluated using PERMANOVA analysis (F-value: 0.9726; R-square: 0.072337; *p* < 0.525) which revealed no significant variation in beta diversity among the experimental groups.

### 3.2. Gut Microbial Compositions and Abundance with Respect to the Healthy Condition and Based on 16S rRNA Amplicon Sequencing

For the subsequent analysis, the annotated ASVs yielded a total of 11 classifiable phyla (Figure 2A) and 85 genera (Figure 2B). Firmicutes, Bacteroides, and Proteobacteria were the top three predominant phyla in the three experimental groups. Additionally, compared to the control group, the abundance rate of observed phyla such as Euryarchaeota, Fusobacteria, Lentisphaerae, and Proteobacteria was increased in the cancer group, followed by the polyps group. The top three dominant bacteria in abundance at the genus level were *Bacteroides*, *Alistipes*, and *Roseburia* associated with the control group, and *Bacteroides*, *Clostridium*, and *Prevotella* in the polyps group, whereas *Bacteroides*, *Clostridium*, and *Butyricimonas* were associated with cancer patients, sequentially according to higher abundance. There were 15 genera observed only in the cancer group, such as *Methanobrevibacter*, *Coxiella*, *Actinomyces*, *Coriobacterium*, *Paraeggerthella*, *Slackia*, *Porphyromonas*, *Vestibaculum*, *Abiotrophia*, *Leuconostoc*, *weissella*, *Peptostreptococcus*, *Bulleidia*, *Cloacibacillus*, and *Haloferula*. Furthermore, seven genera were observed in both the polyps and cancer groups, but not in healthy controls. The abundance of these genera increased or remained the same in the cancer group compared to the polyps group, which included *Rothia*, *Atopobium*, *Coprococcus*, *Anaerotruncus*, *Sporobacter*, *Limnobacter*, and *Acinetobacter*. In comparison with the control group, there were five genera with increased abundance in the cancer and polyps groups, which included, *Eggerthella*, *Butyricimonas*, *Streptococcus*, *Defluviitalea*, and *Ruminococcus*. On the contrary, eight genera, namely, *Bifidobacterium*, *Macellibacteroides*, *Alistipes*, *Odoribacter*, *Paenibacillus*, *Oscillospira*, *Selenomonas*, *WAL_1855D*, showed a decreased abundance in the cancer and polyps group as compared to the control group.

### 3.3. Distribution of Firmicutes and Bacteroides Ratios among Experimental Groups

To further explore the microbial dysbiosis, we checked the Firmicutes and Bacteroides ratio as a marker in experimental groups, as shown in Appendix A. The Firmicutes/Bacteroides ratio was observed to be significantly higher in the cancer group (1.9) followed by the polyps group (1.6) as compared to healthy controls (1.4).

### 3.4. Differential Abundance of Bacterial Diversity among the Experimental Groups

To investigate the statistically significant differences in microbial diversity and abundance at the phylum level in each group (healthy, polyps, and cancer) we used Statistical Analyses of Metagenomic Profiles (STAMP) software using a two-sided White’s non-parametric *t*-test (*p* < 0.05) following a Benjamini–Hochberg analysis to control for false discovery rates. The STAMP statistical comparison is shown in Figure 3, and revealed that only Synergistetes at the phylum level was significantly enriched in the healthy control and polyps groups. The abundance of this bacterium significantly decreased in the polyps group as compared to healthy controls. Similarly, only this bacterium was also found to be significantly enriched in the polyps and cancer groups. The abundance was decreased in the polyps as compared to the cancer group. However, healthy control and cancer groups did not show any significant enrichment.

To further investigate changes in bacterial diversity and abundance at the genus level in healthy, polyps, and cancer groups, the comparison analysis revealed a total of 12 genera were significantly enriched in the cancer and healthy groups, including *Rothia*, *Limnobacter*, *WAL_1855D*, *Coriobacterium*, *Succinatimonas*, *Methabobrevibacter*, *Osllospira*, *Abiotrophila*, *Acinebacter*, *Coxiella*, *Haloferula*, and *Leuconostoc* (Figure 4). Among these significant genera, except for *Oscillosprira* which was highly enriched in healthy controls, all the other 11 significant genera were highly enriched in the cancer group. As shown in Figure 4 left, a comparison of the polyps and cancer group showed a total of 18 significantly enriched genera. Among them *Aeromonas*, *WAL_1855D*, *Oscillospira*, and *Mitsuokella* were in significantly higher abundance in the polyps group, whereas the remaining 14 genera were in significantly higher abundance in the cancer group. A comparison of the healthy controls and polyp group showed six significantly abundant genera. Among them, *Anaerotruncus* and *Mitsuokella* were in significantly higher abundance in the polyps group. However, the abundance of the other four significant genera including *Cytophaga*, *Campylobacter*, *Enterococcus* and *Pseudomonas* showed a decrease in abundance in the polyps as compared to the healthy control group. As shown in Figure 4, based on tumor presence and absence, the abundance of *Olsenella* and *Lactobacillus* at the genus level was significantly reduced in the patient group as compared to healthy controls. However, LEfSe analysis showed no significant difference in the microbial abundance among the experimental groups (Appendix A).

### 3.5. 16S rRNA-Based Community-Level Functional Prediction of Health Conditions

To investigate the microbial functional prediction in health conditions, PICRUSt2 was implied based on 16S rRNA amplicon sequence data. A total of 154 KEGG level 3 functions were identified which generated a total of 32 functions at KEGG level 2. For clear visualization and interpretation at KEGG level22, functions were further analyzed for significant enrichment analysis among the three groups (healthy, polyps, and cancer) using White’s nonparametric *t*-test, and multiple correlations among these significant pathways were performed by Benjamini–Hochberg FDR analysis in STAMP. As shown in Figure 5, a total of 15 pathways were significantly enriched in abundance among healthy, polyps, and cancer groups. Among these, and associated with the polyps and cancer groups, were increases in the abundance of these pathways: metabolism of other amino acids, glycan biosynthesis and metabolism, biosynthesis of other secondary metabolisms, xenobiotic biodegradation and metabolism. Additionally, two pathways related to cancer, namely cancer: overview and cancer: specific types, were increased in the cancer group compared to healthy subjects. Similarly, these pathways were also increased significantly in the polyps compared to the heathy group. However, only the metabolism of other amino acids pathway was significantly enriched in abundance in the polyps as compared to the cancer group.

### 3.6. 16S rRNA-Based Community-Level Functional Profile and Its Correlation between Experimental Groups

A Spearman correlation analysis was performed to further explore the correlation between the microbial community at the genus level and the predicted functions at KEGG level 2, as shown in Appendix A. The statistical significances were considered highly positive and negative at *p* < 0.05 and <0.01, respectively. For better visualization, we only considered statistically significant genera-associated predicted functions. As shown in Figure 6, the correlation analysis revealed that all genera showed a significant correlation with at least one predicted function. Among the positively correlated genera, *Dorea* showed a strongly positive correlation (*p* < 0.01, 0.05) with 87.7% (*n* = 12/14) predicted functions. This was followed by *Ruminococcus* (*p* < 0.01, 0.05) 78.6% (*n* = 11/14), *Clostridium* (*p* < 0.01, 0.05) 78.6% (*n* = 11/14), *Bacteroides* (*p* < 0.01, 0.05) 71.4% (*n* = 10/14), *Eggerthella* (*p* < 0.01, 0.05) 71.4% (*n* = 10/14), *Streptococcus* (*p* < 0.5) 50% (*n* = 7/14), *Atopobium* (*p* < 0.5) 43.9% (*n* = 6/14), *Pseudomonas* (*p* < 0.5) 14.3% (*n* = 2/14), and *Actinomyces* (*p* < 0.5) 14.3% (*n* = 2/14), respectively. Other positively correlated genera showed a significant association with one respective predicted function. Additionally, five genera including *Desulfovibrio*, *Odoribacter*, *Prevotella*, *Scuccinatimonas*, and *Elizabethkingia*, showed a 71.4% (*n* = 10/14), 43.9% (*n* = 6/14), 14.3% (*n* = 2/14), 14.3% (*n* = 2/14), and 7.1% (*n* = 1/14) negatively significant correlation with predicted functions, respectively.

## 4. Discussion

Accumulating evidences have highlighted that the breakdown of gut microbiota is linked to a variety of disorders, such as inflammatory bowel disease, diabetes, obesity, and various types of cancer, particularly colorectal cancer [14,44]. In this study, we found as compared to healthy controls, a higher number of ASVs associated with the cancer group followed by the polyps group, which is in accordance with a previous report [17]. Many studies indicated that the ASV approach is of much higher resolution than the OUT approach, especially in the human microbiota [41]. Additionally, the alpha diversity indices revealed similar patterns of diversity distribution among the three experimental groups, suggesting that the diversity of gut microbes increase following CRC development and progression. However, differences in diversity have been reported in previous studies, which are mainly associated with the nature of sampling or stages of CRC. The elevation of certain phylum in this study, such as Euryarchaeota, Fusobacteria, Lentisphaerae, and Proteobacteria, were higher in abundance in cancer patients or remained the same in polyps patients as compared to healthy controls, which is consistent with previous studies [45,46]. Several studies have highlighted the role of these phylum in the dysbiosis, proinflammation, and progression of CRC [24,45]. Additionally, Fusobacteria is also considered one of the biomarkers for the prognosis of CRC, because it is consistently associated with CRC-associated microbiota [47,48,49]. Immune modulation is assumed to be the key mechanism by which Fusobacterium plays a role in CRC carcinogenesis, these include increased natural killer cell inhibitors and myeloid-derived suppressor, Fap2 and FadA virulence factors, microRNAs and bacterial metabolism [50,51]. Among the top three dominant bacteria in abundance at the genus level, *Bacteroides* was the most dominant in the three experimental groups, whereas the second and third most prevalent genera were different in the cancer (*Clostridium*, and *Butyricimonas*), polyps (*Clostridium*, and *Prevotella*) and healthy control (*Alistipes*, and *Roseburia*) groups. Additionally, in previous reports, a difference in the predominant bacteria at the genus level has been observed, conferred by various factors such as diet, age, alcohol, geographical location and analytical methods [52,53,54]. *Bacteroides* are predominantly found in the human gut as one of the major microbiota and were previously reported to be associated with healthy subjects, polyps, and cancer patients as a dominant bacterium [55]. In previous reports the genus *Prevotella* has been repeatedly associated with diets rich in fiber, whereas *Bacteroides* are associated with diets rich in fat and animal protein [56,57,58]. Moreover, *Prevotella* has also been associated with inflammatory responses in chronic disease [59]. Therefore, further research is needed to know the potential role of this bacterium in inflammatory responses associated with CRC. Additionally, 16S rRNA amplicon sequencing revealed 15 genera were only associated with the cancer experimental group, which included *Methanobrevibacter*, *Coxiella*, *Actinomyces*, *Coriobacterium*, *Paraeggerthella*, *Slackia*, *Porphyromonas*, *Vestibaculum*, *Abiotrophia*, *Leuconostoc*, *weissella*, *Peptostreptococcus*, *Bulleidia*, *Cloacibacillus*, *Haloferula*, indicating these microbes might have a special role in CRC initiation and development. However, the role of the majority of these bacteria in CRC is still scarce. Nowadays, discussion on an increase or shift of the archaeal population and methanogenic bacteria is becoming a hot topic associated with CRC [60,61]. A previous study also indicated a significantly higher abundance of *Methanobrevibacter* in fecal samples was associated with the tumor group as compared to the control group [17]. Recently, the density of the genus *Methanobrevibacter* has been negatively correlated with butyrate concentrations which have been reported to be involved in supplementing energy to epithelial cells and altering mutagenic or toxic compounds [62]. This can support microbiota-related inflammatory events that occur in CRC pathogenesis, due to the fact that methane-producing bacteria consume SCFA [61]. Previous studies have suggested that the higher production of methane could lead to CRC [63,64,65].

The possible reasons for the high unassigned reads of Figure 2 might be due to the difference in the taxonomy of selected classifiers (including reference databases) under the ASV approach. In previous studies, it has been documented that the Greengene database has some limitations for 16S rRNA taxonomic profiling because this database is no longer being maintained [66,67]. For example, in the case of unclassified Kingdom or unclassified phylum taxonomy assignments, the taxonomy classifier was confident enough to assign phylum taxonomy assignments, but not high enough to assign the species or genus or species level, respectively. Regarding the taxonomy classifiers, it could be improved by refinements to the taxonomy classifiers [68,69] and by updating reference databases, such as SILVA or RDP [70,71,72,73]. The cross validation of bacterial taxonomy was performed using the SILVA database (Appendix A), which revealed 181 genera and a reduced ratio of unassigned reads compared to the Greengene database, indicating the low resolution associated with the Greengene database might be due to the low amount of bacterial DNA sequencing data. Recent studies have reported that compared to Greengene and RDP databases, the SILVA database has the highest per-read accuracy and lowest error rates in 16S metagenomic classification, regardless of the software used in classification [70,74]. However, Balvočiūtė and Huson indicated that SILVA, RDP, and Greengenes map well into NCBI, and what kind of database is best for the 16S amplicon survey is based on the research aim [70]. The advantage of Greengene is its species-level identification and high accuracy, but the weakness is its lesser amount of microbial sequence data.

*Firmicutes* and *Bacteroidetes* are the two most commonly dominant bacteria of the human gut and their composition remains relatively unaffected in healthy subjects [75]. However, the ratio of F/B is considered to be potentially correlated with several diseases that can be used as a preliminary basis for diagnosing certain diseases [76]. In this study, the ratio of F/B was found to be higher in patients with colorectal cancer, followed by polyps patients, than in healthy controls, which is in line with previous studies [77,78].

Several studies have demonstrated the shift of the microbial community in CRC patients and healthy controls at different taxa levels [17,79]. In this study there was only one phylum, namely *Synergestetes*, which was significantly enriched in the healthy control and polyps groups, and a decrease in the mean proportion of this phyla was observed in the polyps group. Similarly, it was found significantly enriched in abundance in the cancer and polyps groups, and a decrease in abundance was associated with the polyps as compared to the cancer group. Thus, the significant shift of this bacterium among the experimental groups could be a possible biomarker for the early detection of polyps in CRC patients. A previous study has demonstrated a similar trend where a decreased abundance of *Synergistetes* at the phylum level was observed with the progress of CRC [80].

At the genus level, *Rothia*, *Limnobacter*, *WAL_1855D*, *Coriobacterium*, *Succinatimonas*, *Methabobrevibacter*, *Abiotrophila*, *Actinobacteria*, *Coxiella*, *Haloferula*, and *Leuconostoc* were significantly predominant in the cancer group as compared to healthy controls. However, the mean proportion of Oscillopsia was reduced in the cancer group when compared to healthy controls. These results indicate that these highly enriched bacteria have a strong relationship with cancer and may have a vital role in the development and progression of CRC. In a previous report it is highlighted that a microbiota change indeed promotes the occurrence and development of CRC [80]. The enrichment analysis showed Acinetobacter was the most highly enriched genus in the cancer group. The next generation sequencing of fecal and luminal microbiota of CRC patients and healthy subjects has revealed the higher richness of Acinetobacter at a genus level to be associated with CRC [81]. Similarly a recent study based on 16S rRNA amplicon sequencing has highlighted the envelopment of the genus Acinetobacter in adenomas and diverticula [82]. The genus *Weissella* belong to lactose fermenters, and certain species act as pathogens and occur mostly in patients with impaired host defenses [83]. Additionally, the genus *Rothia* has also been described as a human pathogen [84]. The enrichment of pathogenic bacteria associated with cancer, suggest that they might be involved in the disruption of the intestinal environment by causing pH changes, as previously described in the case of Helicobacter [81].

Additionally, the enrichment comparison of the cancer with the polyps groups revealed 18 significant enriched genera, of which 14 were highly enriched in abundance in cancer, whereas only four genera were highly abundant in the polyps group (Figure 4). These findings suggest that a higher shift of bacteria at the genus level represent their positive role in the development and progression of CRC. In enrichment analysis, the most abundant genus associated with cancer was identified as *Pyramidobacter*, followed by *Peptostreptococcus*, whereas *Aeromonas* was found as the predominant genus associated with polyps. In a previous study on oral microbiota, the role of *Peptostreptococcus* has been described as a biofilm producer which protects the cancerous cells from host immunity [85]. Therefore, the presence of this genus suggests the involvement of *Peptostreptococcus* in CRC progression. *Peptosteptococcus* has been identified to be involved in the proliferation of CRC by inducing the biosynthesis of intracellular cholesterol [86]. *Pyramidobacter* being a sulfidogenic bacteria associated with diets higher in protein intake, can produce H_2_S, which is one of the key factors leading to the impairment of mucus barriers at tumor sites, facilitating tumor-elicited inflammation [87].

The enrichment analysis at the genus level also showed a significant association of six genera in the polyps and healthy control groups. Among these *Cytophaga*, *Campylobacter*, *Pseudomonas*, and *Enterococcus* were highly enriched in abundance and associated with healthy controls, whereas *Anaerotruncus* and *Mitsoukolia* were significantly in abundance and associated with the polyps group. A lower abundance of these bacteria in the polyps as compared to healthy control group suggest the dysbiosis of gut microbiota, as *Enterococcus* and other facultative anaerobes or aerotolerant bacteria are vital for the microbial homeostasis in the large intestine [81]. Similarly, a previous study indicated the higher enrichment of *Enterococcus* and *Anaerotruncus* along with other bacteria in CRC [88].

Additionally, in this study, only two genera, including *Olsenella* and *Lactobacillus*, were found significantly enriched in healthy control and patient groups, based on tumor presence and absence. The abundance of these genera significantly declined in the patient group compared to the healthy control group, which shows that they could be considered a biomarker in the early detection of CRC, and a risk factor of CRC. However, further research is needed to explore these genera up to the species level to make potential biomarkers related with CRC. Both *Lactobacillus* and *Olsenella* are the inhabitants of the human gut and have been previously found in association with reduced gut inflammation [17,89]. Additionally, *Lactobacillus* has been suggested as one of the probiotics to inhibit the development and progression of gut epithelial cells [17]. However, little is known about the role of the *Olsenella* species, including its disappearance during CRC which needs to be further studied.

However, using LEfSe analysis we revealed no significant differences in microbial abundance among the experimental groups. It is a big challenge to identify the differences of two or more phenotypes in any metagenomic dataset [90]. In order to find statistical significance, biological consistency, and the effect size estimation of predicted biomarkers, many approaches or analyses from simple to complex have been established [91]. Regarding simple approaches or analyses, the *t*-test or ANOVA and non-parametric tests, such as the Wilcoxon test, White’s *t*-test, Kruskal–Wallace test, and Welch’s *t*-test, are often used to find statistically significant differences between two groups, which also have the option of adjusting *p*-values using the Benjamini–Hochberg method [91]. Regarding advanced analysis, there is much software constructed to summarize statistical significance, biological consistency, and the effect size estimation of predicted biomarkers, including Metastats/metagenomeSeq, LEFSe, STAMP, etc. [90,91,92]. However, the results of these statistical analysis approaches are all credible, and the most suitable analysis results can be selected according to the associated research background.

Previous studies have shown that various gut microbes and their associated metabolites exhibit proinflammatory and procarcinogenic properties which ultimately exert a great impact on colorectal carcinogenesis [13,14,15]. Recently it has been highlighted that dietary fat, microbially induced lipids and obesity, may lead to major factors that contribute to the increased rate of early-onset CRC [93]. Dysregulation of lipid metabolism in cancer cells, which is increasingly recognized as one of the characteristics of aggressive cancer, correlates with a poorer prognosis and shorter disease-free survival in CRC [93,94]. In this study, the relative abundance of lipid metabolism increased significantly in both the polyps and cancer groups compared to healthy subjects. Similarly, a previous study, based on metabolomics analysis using H NMR spectroscopy, highlighted a significant increase in the level of lipids in the malignant tissues of colon mucosa [30]. Additionally, the KEGG pathway biosynthesis of other secondary metabolites was found to be the second most abundant metabolic pathway. The biosynthesis of other secondary metabolites helps in maintaining the homeostasis of gut microbiota. However, a compromised gut microbiota leads to increased concentrations and the biosynthesis of other secondary metabolites contributing to epithelial-mesenchymal transition, and thus favors the metastatic niche’s setup [95]. Similarly, it has also been reported that xenobiotic metabolism and degradation increase the susceptibility of colonocytes to protumorigenic bacterial metabolites [96]. In the present study, xenobiotic metabolism and degradation was the third most abundant metabolic pathway associated with polyps and cancer groups. Glycosylation is a key post-translational process associated with proteins that alters protein functions and plays a vital role in various biological processes. It has been observed that the aberrant process of glycosylation is linked with the occurrence and progression of different types of cancer, including CRC [97]. A higher abundance of this pathway was significantly associated with polyps and cancer groups than with healthy subjects, suggesting that this is particularly important in the pathology of microbiota-associated diseases such as CRC. Similarly, previously in gut microbiota-associated diseases, such as inflammatory bowel disease, various microbe species have been linked with the metabolism of mucus glycans [98]. The proliferation of cancer cells also needs an abundant supply of amino acids that act as substrates for protein synthesis and are essential in energy generation and the maintenance of cellular redox hemostasis. The nutrient-poor microenvironment of cancer and stromal cells requires a relationship with the microbial community to fulfill their essential nutrients [99]. This study found a higher abundance of amino acid metabolism associated with polyps and cancer groups. Additionally, two cancer-related KEGG pathways, such as cancer overview and specific types of cancer, were in higher abundance in polyps and cancer groups compared with healthy subjects. At KEGG level 3, these two pathways belonged to choline metabolism in cancer, colorectal cancer, and MicroRNAs in cancer (KEGG level 3 not shown here). In previous studies, it has been highlighted that an increased abundance of choline is responsible for the signal transduction or growth promotion of various human malignancies, including colon cancer and adenoma [100,101]. Similarly, a previous study, based on metabolomics analysis using H NMR spectroscopy, highlighted a significant increase in the level of choline in the malignant tissues of colon mucosa [30]. Additionally, MicroRNA is a short endogenous RNA molecule that regulates post-transcriptional gene expression and is considered a key factor for the mutagenesis of CRC, leading to a potential biomarker for the diagnosis, prognosis, and therapy of CRC [102]. This study’s overall functional prediction associated with gut microbiota, suggests that the metabolic reprogramming and these microbes’ metabolic products may serve as potential biomarkers for the increased risk of CRC development and progression in humans. However, further confirmation of the presence of these microbial metabolites needs proteomic based analysis.

A Spearman correlation analysis was performed to examine the correlation between the predicted functions and microbial community of the three experimental groups. The analysis revealed that *Dorea*, *Ruminococcus*, *Clostridium*, *Bacteroides*, *Eggerthella*, *Streptococcus*, and *Atopobium* were strongly positive in their correlation with the majority of the predicted functions, indicating these microbes can independently play an important role in the initiation and progression of CRP. Several studies have highlighted that the metabolic contribution of these microbes is associated with the development of CRC. For example, the genus *Dorea* has the capability to adhere to cancerous cells and may confer this genus a competitive benefit in the cancerous colorectal environment [103]. The increased level of *Ruminococcus* metabolites associated with lipid and amino acid metabolism have been detected from the stool samples of CRC patients [104]. The role of these significantly positive bacteria in xenobiotic biodegradation and metabolism by their enzymes, have also been highlighted in previous studies [105]. Additionally, *Desulfovibrio* showed a significantly negative correlation with the majority of predicted functions, followed by *Odoribacter*, *Prevotella*, *Scuccinatimonas*, and *Elizabethkingia*, suggesting these bacteria are involved in the regulation of their respective predicted functions.

The overall taxonomic findings suggest that the significant shift of microbial diversity increases with the progress of CRC. For a more clear understanding we can say that the shift of microbes was less in polyps patients and higher in cancer patients when comparing them with the healthy controls. The shift in microbial taxa at different taxonomic levels could be a possible biomarker in the diagnosis and treatment of CRC and needs further extensive validation. However, the role of gut microbiota and their community structure in different precursors of CRC have not been clearly discussed in previous studies. Additionally, the functional predictions associated with gut microbiota suggest that the metabolic reprogramming of their metabolic products may serve as potential biomarkers for the increased risk of CRC development and progression at different stages in humans. However, for further confirmation of the presence of these microbial metabolites we suggest that the use of metabolomic and proteomic based analyses might be considered ideal methods.

## Figures and Tables

**Figure 1 microorganisms-09-01548-f001:**
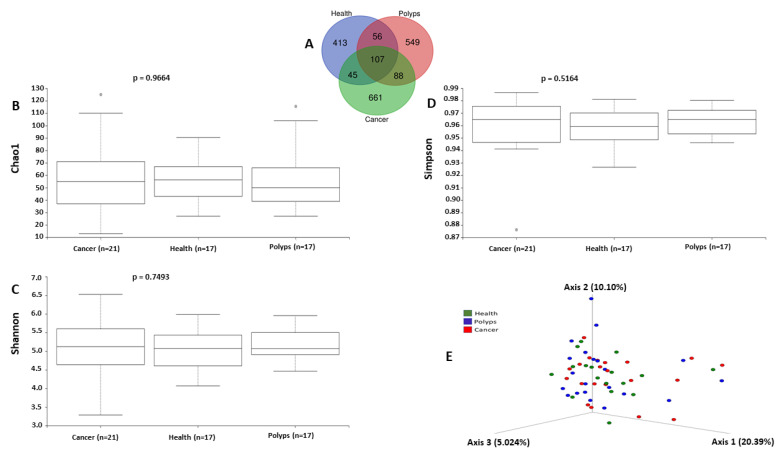
Venn diagram (**A**) representing the shared and unique ASVs among the three experimental groups. Boxplots showing alpha diversity ((**B**) Chao1, (**C**) Shannon, and (**D**) Simpson) among the experimental groups. PCoA (**E**) showing the bacterial beta diversity among the experimental groups based on weighted UniFrac distances.

**Figure 2 microorganisms-09-01548-f002:**
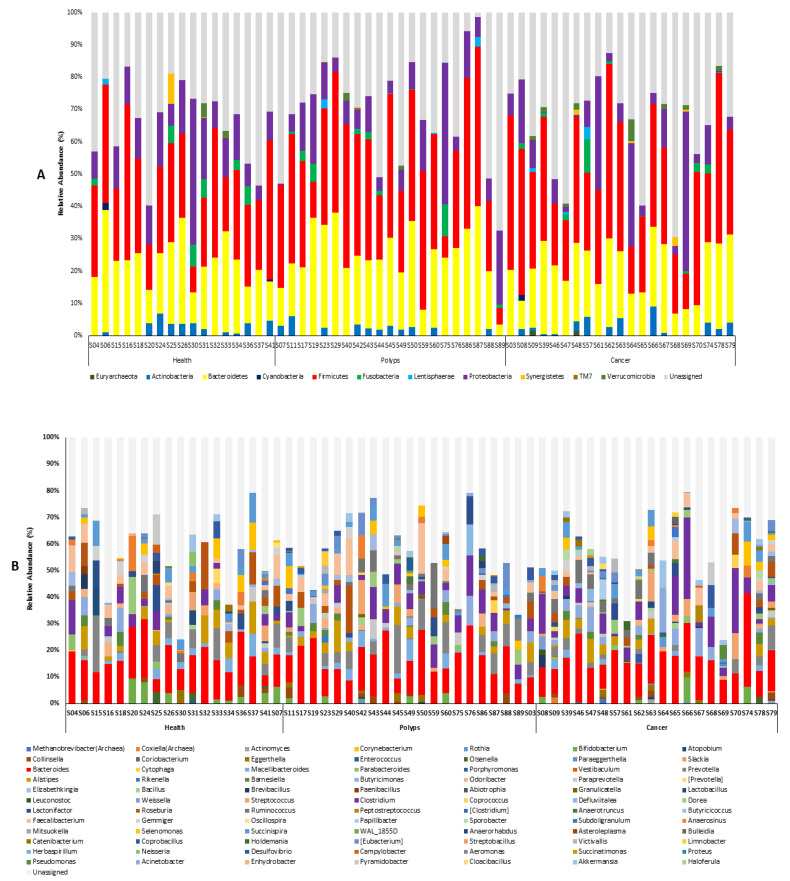
16S rRNA amplicon sequencing-based relative abundances of microbial diversity at the phylum level (**A**) and genus level (**B**) in each experimental group.

**Figure 3 microorganisms-09-01548-f003:**
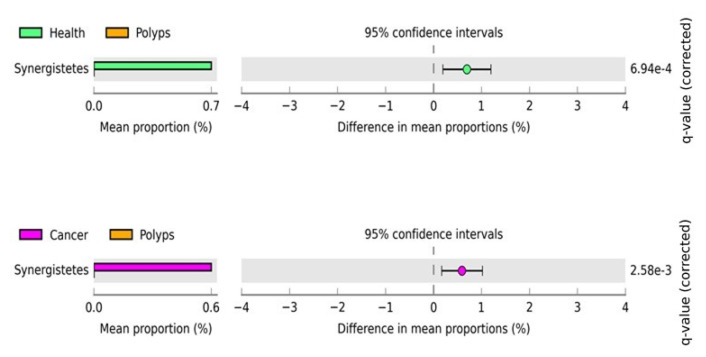
The post hoc plot of enriched bacterial phyla among three health conditions (healthy, polyps, and cancer). The left side of these figures shows the abundance ratio of differentially enriched bacterial phyla. The right side represents the significant difference at *p* < 0.05, whereas the middle one indicates the mean proportion of differentially enriched bacterial phyla in the 95% confidence.

**Figure 4 microorganisms-09-01548-f004:**
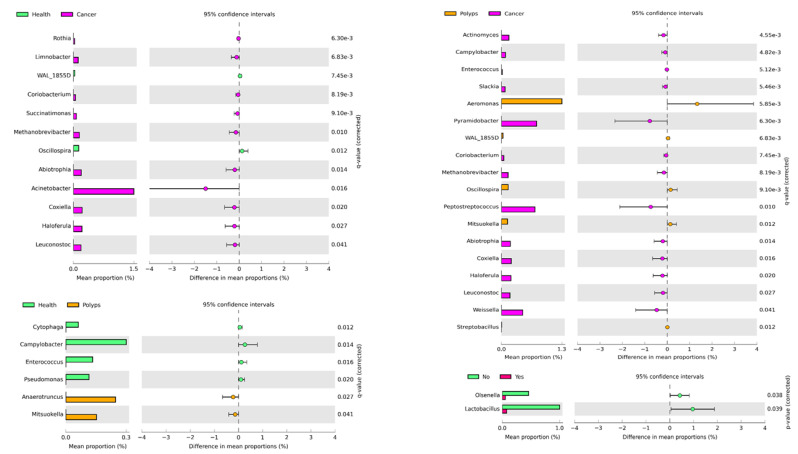
The post hoc plots of enriched bacterial genera among three health conditions (healthy, polyps, and cancer). The left side of these figures shows the abundance ratio of differentially enriched bacterial genera. The right side represents the significant difference at *p* < 0.05, whereas the middle one indicates the mean proportion of differentially enriched bacterial genera in the 95% confidence interval.

**Figure 5 microorganisms-09-01548-f005:**
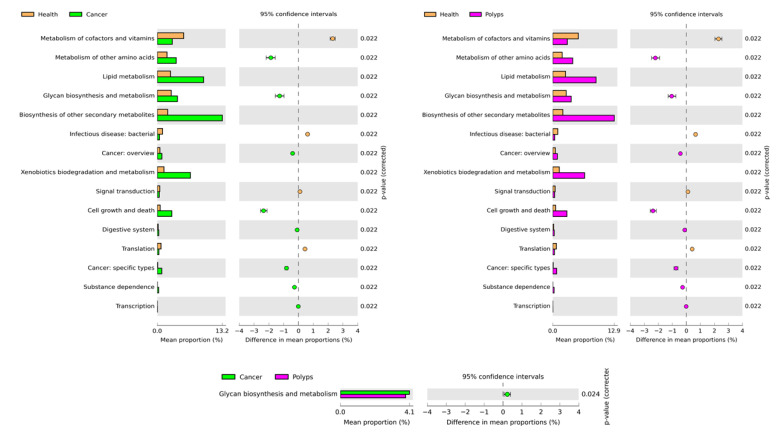
The post hoc plot of enriched microbial predicted functions among three health conditions (healthy, polyps, and cancer). The left side of these figures shows the abundance ratio of differentially enriched KEGG pathways. The right side represents the significant difference at *p* < 0.05, whereas the middle one indicates the mean proportion of differentially enriched KEGG pathways in the 95% confidence.

**Figure 6 microorganisms-09-01548-f006:**
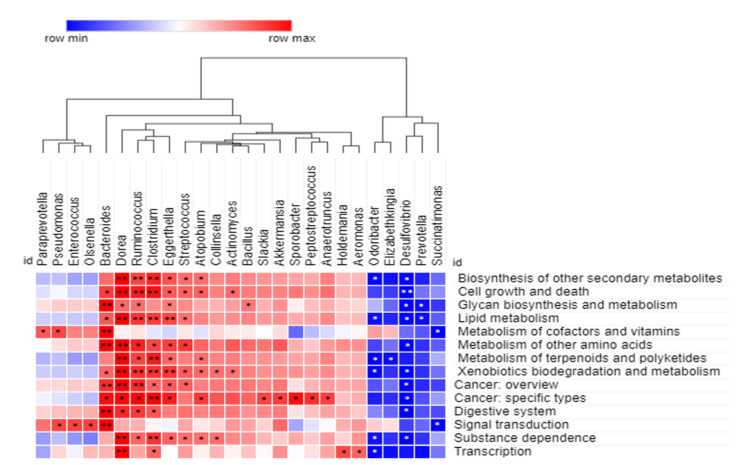
Correlations between the bacterial experimental groups community and predicted functional profiles, based on 16S rRNA amplicon sequencing. Spearman correlation analysis heatmap constructed for each pairwise comparison between level 2 Kyoto Encyclopedia of Genes and Genomes (KEGG) pathways and bacterial taxa at the genus level. The positive and negative correlations are indicated in red and blue colors, respectively. The correlation was considered significant at ** *p* < 0.01 and * *p* < 0.05.

## Data Availability

The datasets generated during and/or analyzed during the current study are available from the corresponding author on reasonable request.

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
