# Peer review of "Colorectal Cancer Stage-Specific Fecal Bacterial Community Fingerprinting of the Taiwanese Population and Underpinning of Potential Taxonomic Biomarkers"

_microorganisms, 2021, doi:10.3390/microorganisms9081548_

Round 1

Reviewer 1 Report

In this manuscript, Chuan-Yin Fang and colleagues investigated the gut microbial profile in healthy, polyps, and colorectal cancer patients using the 16S rRNA amplicon sequencing. The authors compared the microbial structure and composition, as well as predicted functions in the gut microbiota. While the manuscript provided readers with valuable information about bacterial taxa identified in colorectal cancer, however, a lack of sufficient details about methodology compromised the message of the study. Issues also need to be clarified for further improvement with respect to statistical analysis and data interpretation.

Specifically,

  1. Please add line numbers for review.
  2. Please change the wording to be more accurate. (For example, please change the term 16S metagenomic analysis/metagenomic function to 16S rRNA amplicon sequencing; please avoid using the term gut flora).
  3. Regarding the subjects included in the current study, please provide information about their dietary pattern as diet plays a major role in shaping gut microbial communities.
  4. Was a bead-beating step included in the DNA extraction? Was there a positive control used in the sequencing process to indicate the efficiency of DNA extraction and sequencing?
  5. Please add more information about the sequencing analysis. For instance, which denoising method was used (dada2 or deblur)? What was the truncation size? Please present the average read number. Were the reads rarefied for diversity analyses? What about the results using the Bray-Curtis similarity matrix?
  6. There were a significant proportion of unassigned reads (up to 50%). Please provide an explanation and discussion.
  7. Regarding the statistical analysis, please perform PERMANOVA analysis such as adonis to investigate the differences in beta-diversity. Was there any reason to perform the non-parametric t-test only as there were three treatment groups? Please add P values and significance labels to figures.
  8. For biomarker identification, it is highly recommended to use Linear discriminant analysis Effect Size (LefSe) to validate the results.
  9. The way to present bacterial taxa with its proportion could be quite confusing to readers (Rothia (0%, 4.8%, 4.8%)). Please revise accordingly. Providing a table with detailed values might be a good option.

Author Response

Review #1 comments

English language and style

English language and style

( ) Extensive editing of English language and style required
(x) Moderate English changes required
( ) English language and style are fine/minor spell check required
( ) I don't feel qualified to judge about the English language and style

In this manuscript, Chuan-Yin Fang and colleagues investigated the gut microbial profile in healthy, polyps, and colorectal cancer patients using the 16S rRNA amplicon sequencing. The authors compared the microbial structure and composition, as well as predicted functions in the gut microbiota. While the manuscript provided readers with valuable information about bacterial taxa identified in colorectal cancer, however, a lack of sufficient details about methodology compromised the message of the study. Issues also need to be clarified for further improvement with respect to statistical analysis and data interpretation.

Response: We thank the reviewer for the encouragement. We have carefully checked the grammar and revised the manuscript. Furthermore, we have decided to get the MDPI English editing service after the completion of revision or when reviewer ask for editing at any time.

Comments:

Please add line numbers for review.

Responses: Thanks for the reviewer’s comment. We have added the line numbers as suggested.

Please change the wording to be more accurate. (For example, please change the term 16S metagenomic analysis/metagenomic function to 16S rRNA amplicon sequencing; please avoid using the term gut flora).

Responses: Thanks for the reviewer's comments. We have changed the description throughout this manuscript as suggested.

Regarding the subjects included in the current study, please provide information about their dietary pattern as diet plays a major role in shaping gut microbial communities.

Responses: Thanks for the reviewer’s comment. The present study focused only on microbial dysbiosis associated with CRC. The dietary pattern will be considered in the future as this project is still under process.

Was a bead-beating step included in the DNA extraction? Was there a positive control used in the sequencing process to indicate the efficiency of DNA extraction and sequencing?

Responses: Thanks for the reviewer's comment. Yes, the protocol of the DNA extraction kit contains a bead-beating step. We did not use any control in the sequencing process to check DNA extraction or sequencing efficiency. Instead, we used Nanodrop and gel electrophoresis to check the quality and quantity of extracted gDNA as already described in the methodology section.

Please add more information about the sequencing analysis. For instance, which denoising method was used (dada2 or deblur)? What was the truncation size? Please present the average read number. Were the reads rarefied for diversity analyses? What about the results using the Bray-Curtis similarity matrix?

Responses: Thanks for the reviewer’s comments. We have added more information in the methodology and result section as suggested (Lines: 183-189, 230-233).

There were a significant proportion of unassigned reads (up to 50%). Please provide an explanation and discussion.

Responses: Thanks for the reviewer’s comments. We have added a description as suggested in the discussion section (Lines: 412-421). We used the Greengene database as a taxonomic classifier that did not provide enough resolution to classify all ASV to genus level due to limitations in this database as it has not been updated for a long time. We specifically used this database because we have to use PICRUSt2 analysis to predict metabolic function, which is only possible through the Greengene database. 

Regarding the statistical analysis, please perform PERMANOVA analysis such as adonis to investigate the differences in beta-diversity. Was there any reason to perform the non-parametric t-test only as there were three treatment groups? Please add P values and significance labels to figures.

Responses: Thanks for the reviewer’s comment. We have added a description in the result section along with the p-value as suggested (line: 230-233). We used non-parametric tests because of the nature of metagenomic data. Relative abundances, in most cases, violate the main assumption of typical parametric tests (normal population in each class), whereas non-parametric tests are much more robust to the underlying distribution of the data since they are distribution-free approaches.

For biomarker identification, it is highly recommended to use Linear discriminant analysis Effect Size (LefSe) to validate the results.

Response: Thanks for the reviewer’s comment. As per your recommendation we considered the Lefse analysis (Lines: 194-197, 306-307, 492-505).

The way to present bacterial taxa with its proportion could be quite confusing to readers (Rothia (0%, 4.8%, 4.8%)). Please revise accordingly. Providing a table with detailed values might be a good option.

Responses: Thanks for the reviewer’s comment. We have changed the description as suggested to make it more understandable for the readers (Lines: 244-262).

Reviewer 2 Report

The manuscript brings new information about the faecal microbial community and functionality by analysing faecal samples from healthy, polyps and cancer groups. It is well written, the text has an excellent rational flow, and I recommend this manuscript for publication after minor revisions.

Comments:

- Figure 1: the axis legend of graphs B, C and D are too small (and it also applies for figure 2), and figures B and C have remained "health" on the x-axis.

- Topic 3.2: it is confusing how the three groups' percentage of the abundance inside parenthesis is showed. If you are always using the same order of groups (control, polyps and cancer), I suggest you state it at the begging of topic 3.2 and use this same pattern on Figures 1 B, C and D. 

- For me, it was strange the way the legends of figures 3, 4 and 5 are written because of the usage of the terms "left", "right" and "middle". Since there is no clear "left", "right" and "middle", and you have more than one graph in each figure.

- Results text from figure 3: I would suggest adding the information that only significant results are showed in the figure. And, it is stated, "The STAMP statistical comparison is shown in Fig. 3, which revealed that only Synergistetes at phylum level was significantly enriched in healthy control and polyps group." However, fig. 3 is showing enrichment in control and cancer groups.

- Results text from figure 3: "The abundance of this bacterium significantly decreased in polyps as compared to healthy controlSimilarly, only this bacterium was found significantly enriched in polyps and cancer group. The abundance was decreased in polyps as compare to the cancer group. However, there was no any significant enrichment between healthy control and cancer group." This part of the text is confusing, and I suggest you rewrite it.

- Results text from figure 4: "Comparison of polyps and cancer group showed a total of 18 significantly enriched genera as shown in Fig. 5." It is figure 4.

- Topic 3.5: "A total of 15 pathways were significantly enriched in abundance among health, polyps, and cancer groups as shown in Fig. 4." Here, I believe is figure 5. 

Figure 5: Why is there no simbol represented in the 95% confidence intervals for some KEGG pathways? For example, "Lipid metabolism" and "Biosynthesis of other secondary metabolites".

- Figure 6 legend: change green for blue colour in the description.

- Supplementary table 2 is missing.

Author Response

Review #2 comments

English language and style

( ) Extensive editing of English language and style required
( ) Moderate English changes required
(x) English language and style are fine/minor spell check required
( ) I don't feel qualified to judge about the English language and style

The manuscript brings new information about the faecal microbial community and functionality by analysing faecal samples from healthy, polyps and cancer groups. It is well written, the text has an excellent rational flow, and I recommend this manuscript for publication after minor revisions.

Comments:

- Figure 1: the axis legend of graphs B, C and D are too small (and it also applies for figure 2), and figures B and C have remained "health" on the x-axis.

Response: Thanks for the reviewer’s comments. We have changed the quality of the figures as suggested (Lines: 235, 263, and 264).

Topic 3.2: it is confusing how the three groups' percentage of the abundance inside parenthesis is showed. If you are always using the same order of groups (control, polyps and cancer), I suggest you state it at the begging of topic 3.2 and use this same pattern on Figures 1 B, C and D. 

Response: Thanks for the reviewer’s comment. We have changed the description as suggested to make it more understandable for the readers (Lines: 244-262).

- For me, it was strange the way the legends of figures 3, 4 and 5 are written because of the usage of the terms "left", "right" and "middle". Since there is no clear "left", "right" and "middle", and you have more than one graph in each figure.

Response: Thanks for the reviewer’s comments. Our intended meaning relating to left and right, and middle implies for each figure. Left means the figure part, which denotes mean proportion, the middle part of the figure means the difference in mean proportions, and the right means the end of each figure where the q value is denoted.

Results text from figure 3: I would suggest adding the information that only significant results are showed in the figure. And, it is stated, "The STAMP statistical comparison is shown in Fig. 3, which revealed that only Synergistetes at phylum level was significantly enriched in healthy control and polyps group." However, fig. 3 is showing enrichment in control and cancer groups.

Response: Thanks for the reviewer’s comments. We have changed the description to make it more understandable as suggested (Lines: 276-282).

Results text from figure 3: "The abundance of this bacterium significantly decreased in polyps as compared to healthy controlSimilarly, only this bacterium was found significantly enriched in polyps and cancer group. The abundance was decreased in polyps as compare to the cancer group. However, there was no any significant enrichment between healthy control and cancer group." This part of the text is confusing, and I suggest you rewrite it.

Response: Thanks for the reviewer’s comments. We have changed the description to make it more understandable as suggested (Lines: 273-282).

- Results text from figure 4: "Comparison of polyps and cancer group showed a total of 18 significantly enriched genera as shown in Fig. 5." It is figure 4.

Response: Thanks for the reviewer’s comments. We have changed the figure number in the description as suggested (Line: 295).

Topic 3.5: "A total of 15 pathways were significantly enriched in abundance among health, polyps, and cancer groups as shown in Fig. 4." Here, I believe is figure 5. 

Response: Thanks for the reviewer’s comments. We have changed the figure number in the description as suggested (Line: 320-322).

Figure 5: Why is there no simbol represented in the 95% confidence intervals for some KEGG pathways? For example, "Lipid metabolism" and "Biosynthesis of other secondary metabolites".

Response: Thanks for the reviewer’s comments. We have set up the minimum and maximum limit of mean difference -4 and 4. That is why the higher or lower value cannot be visible in the figure as in lipid metabolism and biosynthesis of other secondary metabolites.

- Figure 6 legend: change green for blue colour in the description.

Response: Thanks for the reviewer’s comment. We have changed it as suggested (Line: 361).

- Supplementary table 2 is missing.

Response: Thanks for the reviewer’s comment. We have added the missing table in the supplementary file (Named Supplementary table 1).

Reviewer 3 Report

In the current manuscript authors have provided significant insight into usage of  gut microbiota dysbiosis as a potential biomarkers for early diagnosis and treatment of CRC. Result of the current investigation have significant advanced our understanding of shift of microbial diversity in the progression of CRC. The study design is straightforward, and it is a nicely written article with sufficient data to support their research thesis. I have few comments

In introduction section, 2nd paragraph the repeated statement, “The gut mi-crobiota performs a variety of crucial functions that help the host in protecting from path-ogens, nutrient supply, immune modulation, and shaping the intestinal epithelium (Cheng et al. 2020, Kang and Martin 2017).”, should be deleted.

More introduction and discussion should be included regarding FMT or transfer of stool from patients with CRC to germ-free mice alters intestinal cell proliferation and induction on tumour formation. Relevant articles should be cited.

Author has stated Fusobacteria, Bacteroides as potential the biomarkers for CRC. More discussion should be provided on how these bacteria induce colorectal carcinogenesis by alteration of cell signaling pathways.

Author should follow the MDPI standard refence format.

Section 2.1 “Selection of healthy controls and cancer patients” in materials and methods group should be changed to “Patient characteristics” or “Participants”

Lacking of human IRB approval in “Selection of healthy controls and cancer patients” in materials and methods section.

ASV, PCA terms should be abbreviated.

Boxplots in Figure 1 should be colored for better data visualization and interpretation.

Figure axis legends are difficcult to read in some figures. High resolution figures should be provided and font size for figure x and y-axis legends should be increased.

In Figure 2, the bar color should be changed in “unassigned” group in the relative abundance graph for 16S metagenomic relative abundance of microbial diversity.

Supplementary Fig.1. “Figure S1. Ratio of Firmicutes to Bacteroides in experimental groups.” should be included in separate file.

In Supplementary Fig.1. “Firmicutes/Bacteroides ratio was observed to be significantly higher in cancer (1.9) followed by polyps (1.6) as compared to healthy control (1.4). However, Firmicutes/Bacteroides ratio in Supplementary Fig.1. was found to be higher in polyps than cancer. Author should modify the statements in result section.

Result section 3.4, the subtitle  “A shift in microbial diversity with respect to health conditions, and tumor presence/absence’’ should be modified

In Fig.4, the figure axis legends Yes or no in mean proportion % graph for the Olsenella and Lactobacillus at genus level, should be presented as health or cancer.

supplementary table 2. is missing in manuscript.

Author Response

Review #3 comments

English language and style

( ) Extensive editing of English language and style required
(x) Moderate English changes required
( ) English language and style are fine/minor spell check required
( ) I don't feel qualified to judge about the English language and style

In the current manuscript authors have provided significant insight into usage of gut microbiota dysbiosis as a potential biomarkers for early diagnosis and treatment of CRC. Result of the current investigation have significant advanced our understanding of shift of microbial diversity in the progression of CRC. The study design is straightforward, and it is a nicely written article with sufficient data to support their research thesis. I have few comments.

Response: We thank the reviewer for the encouragement. We have carefully checked the grammar and revised the manuscript. Furthermore, we have decided to get the MDPI English editing service after the completion of revision or when reviewer ask for editing at any time.

In introduction section, 2nd paragraph the repeated statement, “The gut mi-crobiota performs a variety of crucial functions that help the host in protecting from path-ogens, nutrient supply, immune modulation, and shaping the intestinal epithelium (Cheng et al. 2020, Kang and Martin 2017).”, should be deleted.

Response: Thanks for the reviewer’s comments. We have changed the description as suggested (Lines: 57-60).

More introduction and discussion should be included regarding FMT or transfer of stool from patients with CRC to germ-free mice alters intestinal cell proliferation and induction on tumour formation. Relevant articles should be cited.

Response: Thanks for the reviewer’s comments. We have added a description related to mice models in the introduction as suggested (Lines: 75-77). Additionally, in the introduction and discussion section, we have already provided the possible mechanisms for gut microbes in the development and progression of CRC. However, much detail about the mice model was not discussed because this study does not focus on the mechanisms.

Author has stated Fusobacteria, Bacteroides as potential the biomarkers for CRC. More discussion should be provided on how these bacteria induce colorectal carcinogenesis by alteration of cell signaling pathways.

Response: Thanks for the reviewer’s comments. We have added a description in the discussion section as suggested (lines: 380-383). 

Author should follow the MDPI standard refence format.

Response: Thanks for the reviewer’s comment. We have changed the format as suggested.

 Section 2.1 “Selection of healthy controls and cancer patients” in materials and methods group should be changed to “Patient characteristics” or “Participants”

 Response: Thanks for the reviewer’s comment. We have changed the description as suggested to make it more understandable.

Lacking of human IRB approval in “Selection of healthy controls and cancer patients” in materials and methods section.

Response: Thanks for the reviewer’s comment. We have added the IRB approval at the end of this manuscript (lines: 586-588)

ASV, PCA terms should be abbreviated.

Response: Thanks for the reviewer’s comment. We have abbreviated these words as suggested (lines: 598-602).

Boxplots in Figure 1 should be colored for better data visualization and interpretation.

Response: Thanks for the reviewer’s comment. We used the QIIME2 NGS processing pipeline, where these kinds of plots are supported without color. For better visualization and interpretation, we have improved the quality of these figures (Line: 235).

Figure axis legends are difficcult to read in some figures. High resolution figures should be provided and font size for figure x and y-axis legends should be increased.

Response: Thanks for the reviewer’s comment. We have improved the resolution of these figures as suggested to make them more readable (Line: 235).

In Figure 2, the bar color should be changed in “unassigned” group in the relative abundance graph for 16S metagenomic relative abundance of microbial diversity.

Response: Thanks for the reviewer’s comment. We used this light color merely to the unassigned group for making other vital groups more visible and prominent. 

Supplementary Fig.1. “Figure S1. Ratio of Firmicutes to Bacteroides in experimental groups.” should be included in separate file.

Response: Thanks for the reviewer’s comment. We have included this figure in the supplementary files.

In Supplementary Fig.1. “Firmicutes/Bacteroides ratio was observed to be significantly higher in cancer (1.9) followed by polyps (1.6) as compared to healthy control (1.4). However, Firmicutes/Bacteroides ratio in Supplementary Fig.1. was found to be higher in polyps than cancer. Author should modify the statements in result section.

Response: Thanks for the reviewer’s comment. If we are not wrong, the description in the result section is according to the figure.

Result section 3.4, the subtitle  “A shift in microbial diversity with respect to health conditions, and tumor presence/absence’’ should be modified

Thanks for the reviewer’s comment. We have changed the subtitle as per suggested (Line: 272).

In Fig.4, the figure axis legends Yes or no in mean proportion % graph for the Olsenella and Lactobacillus at genus level, should be presented as health or cancer.

Thanks for the reviewer’s comment. Other differential significance analysis is based on the healthy controls and two stages of CRC, such as polyps and cancer. However, two groups were made based on tumor yes and no. In tumor “yes” both polyps and cancer are included, whereas in tumor “no” only healthy control is included. 

supplementary table 2. is missing in manuscript.

Thanks for the reviewer’s comment. We have provided in the supplementary files.

Round 2

Reviewer 1 Report

With the revision, a number of concerns have been addressed, particularly regarding sequencing analysis. I have a few issues that need to be further clarified.

Specifically:

  1. With respect to the significant proportion of unassigned reads, it is highly recommended to also check profiles using the SILVA database to rule out possible issues with the sequencing procedure. What is the ratio of unassigned reads from an analysis using the SILVA database? The PICRUST2 is not restricted to the Greengene database. The explanation for these unassigned reads is not satisfied since unassigned reads were not likely due to the resolution as reflected by the phylum level.
  2. L 88-90 – The phylum name should not be italicized. Please check through the manuscript.
  3. L 143 – Please add a bead-beating step to the methodology as the manual from the QIAamp DNA Stool Mini Kit does not include a bead-beating step.
  4. L 233 – Please add the PERMANOVA analysis procedure to the methodology section.
  5. L 281 - Figure 2B – Please check the genus “human”.
  6. L 363 - Please use a consistent format of “p-value”.
  7. L 599 – “Pearson correlation”?

Author Response

Respected Reviewer,

Please find the attached doc file

Reviewer 3 Report

In the current manuscript authors have provided significant insight into usage of  gut microbiota dysbiosis as a potential biomarkers for early diagnosis and treatment of CRC. Result of the current investigation have significant advanced our understanding of shift of microbial diversity in the progression of CRC.  

Minor comments

In the Supplementary Fig. 2. , respective experimental groups should be presented in the cladogram for clear  oresentaion of  bacterial clades.

Author Response

Respected Reviewer,
